# Agricultural Livelihood Types and Type-Specific Drivers of Crop Production Diversification: Evidence from Aral Sea Basin Region

**Akmal Akramkhanov** [1,*] **, Adkham Akbarov** [1] **, Shakhzoda Umarova** [1] **and Quang Bao Le** [2]

1  International Center for Agricultural Research in the Dry Areas (ICARDA), Street Osiyo 6/107, Tashkent 100084, Uzbekistan
2  International Center for Agricultural Research in the Dry Areas (ICARDA), 2 Port Said, Victoria Sq., Ismail El-Shaaer Building, Maadi, Cairo 11728, Egypt
*  Correspondence: a.akramkhanov@cgiar.org

**Abstract:** Understanding the factors driving the farmers' decisions to diversify their crop production is important for management strategies and policies promoting climate-smart agricultural development. Options for diversification and its associated drivers might be shaped by livelihood context, and it remains as a general gap in knowledge. This study aimed to reveal the driving factors behind households' decisions to diversify their crops in different livelihood contexts. This information could be useful to inform stakeholders on a set of context-fitted options for improving natural resources and rural livelihood resilience to climatic variability and risks. This study applied the Sustainable Livelihood Framework (SLF) to guide surveys and multivariate analyses that identified agricultural livelihood context types at the village level, and also evaluated both the common and type-specific drivers encouraging households to diversify their agricultural production in two rural villages in the Aral Sea region. This study objectively identified three distinct agricultural livelihood types and the main factors differentiating these types from each other. When the total sampled population was analyzed, the results indicated that the agricultural experience of the household heads, levels of education, sources of income, number of cattle and land endowments, and proximity to markets were common and significant drivers in diversifying these households' crop production. Analyzing the decisions behind diversifying crop production for each agricultural livelihood type revealed type-specific drivers of diversification. The findings suggested that considering both common and type-specific drivers of diversification would allow better understanding of household decisions and provide more insights to develop effective policies promoting climate-smart agriculture through diversification, rather than continuing to use the current "uniform blanket" approach.

**Keywords:** agricultural livelihood system; smallholder; household; diversification; Karakalpakstan; Aral Sea Basin Region

## 1. Introduction

Agricultural diversification has been widely recognized as one of the major adaptation strategies for sustaining rural livelihoods in the face of climate and other unexpected changes [1–4]. Crops increase productivity and enhance the stability of household livelihoods (e.g., stable household income and food security) [5] and the ecological services of agroecosystems (e.g., improved soil fertility, agro-biodiversity, and reduced emission of greenhouse gasses) [6]. For smallholder farmers, production diversification is one of the most feasible and cost-effective ways of minimizing uncertainties and risks [4,7]. The diversification of production on farms and land management measures are important elements for improving ecological and livelihood resilience at the village level [2,8].

Despite ample evidence of the benefits of agricultural diversification (including genetic resources and management practices) to agricultural production [2,3], natural resources

and rural livelihoods, farmers' decisions regarding agricultural diversification have not been well understood. Adequate understanding of the social, economic and ecological drivers of smallholders' diversification strategies are the key for rural policy makers and developers to improve agricultural and livelihood resilience in rural areas. So far, most of the studies about smallholders' adoptions of diversification options have considered the farmers' choices of particular land types or improved agricultural practices rather than production diversification, which has only been considered by a few recent studies [9].

Another important limitation in many of the previous adoption studies was that the effects of the social, economic, and ecological drivers on the farmers' decisions were the same across their studies of agrarian communities despite the considerable diversity of social-ecological contexts [10,11]. Diversity of social-ecological contexts often occurs even at the village level and can shape the relationships between the farmers' decisions and the relevant drivers. The affecting magnitudes (how strong) and the directions (positive or negative) of common drivers (education, farm size) on household behaviors can differ among households with different contextual livelihood types [12,13]. There have been hardly any studies in current literature that have examined how livelihood context types shape the way (magnitude and direction) socio-economic and biophysical drivers affect farmers' decisions on agricultural diversification.

Insight into the households' strategies for agricultural diversification in different, specific livelihood contexts has often been skipped in numerous interventions to ease the Aral Sea crisis in Central Asia. A study by Fabian Löw (2022) described how the Aral Sea, which was the fourth largest inland lake, has been transformed into Aralkum. Water surface area has been decreased by 90 percent. Consequently, thousands of people residing near the Aral Sea lost their main income sources. Therefore, it is extremely important to pay attention to the livelihoods of the rural populations in the Aral Sea basin (ASB), particularly those residing near the former sea because of their high vulnerability to environmental shocks [14,15]. The small formal employment markets help to mitigate the negative impacts of environmental shocks. However, the small formal employment markets usually exist in rural areas [16], particularly in remote locations of the ASB, and their growth scope is limited. Therefore, the importance of household agricultural production in the ASB has become a vital livelihood option during last decades [17].

Several programs designed to support the agricultural restructuring of, and the livelihood transformations in, the focus region were carried out, including a long-term research program (2002–2011) conducted by the University of Bonn [18], more recent programs jointly conducted by UN organizations [19], and the CGIAR research program on Dryland Systems (CRP-DS) [20,21]. Studies conducted in the ASB indicated that geographical factors affect the agricultural livelihoods of households and their abilities to respond to environmental changes [17], and that diversification is the only secure element of a viable livelihood strategy in such ecosystems [15,22].

Crop choice and livestock production strategies by rural households could help people to cope with not only with environmental challenges, but also with poverty and improving rural people's nutrition. Crop production is severely affected by climatic variability [14], leading to high crop losses and making it difficult for rural households to escape poverty and increase nutrition. Hence, identifying the drivers behind rural households' decisions to allocate land for agricultural production can be used to create future policies that will efficiently improve rural lives.

Much research was carried out to improve the resilience of agricultural systems. However, most of the approaches are one-dimensional, often addressing mainly either biophysical, or technological, economic, or social aspects. Meanwhile, research communities understood that the issues being addressed are intricately interlinked and need to have an interdisciplinary approach. Thus, the system has been deemed complex and requires a different approach. Studies of the most recent CRP-DS programs also recognized that dryland development on the scale envisioned by the Sustainable Development Goals (SDGs) requires researchers to embrace complexity, diversity, and uncertainty, and to look

across different system components, scales, and types of knowledge [21]. As [23] correctly justified, a systematic approach is needed to address the complexity of agricultural and livelihood systems because of the difficulty of developing innovations and interventions that account for such complexity.

The current research builds upon the CRP-DS program interventions and was motivated by approaches described by [9], which utilized the Sustainable Livelihoods Framework (SLF) to investigate the factors that affect agricultural livelihoods at the village level in the ASB, and to determine the type-specific factors in the households' productions and livelihood diversifications compared to the treatment of all the households as a group. The study has been initiated within the framework of the CGIAR research program on Dryland Systems [20].

This study attempted to define and characterize clusters of smallholders' livelihood systems and contexts in the ASB through functional livelihood typologies that provided better targeting in system research and development and up- and out-scaling of place-based findings. This study's objectives were to reveal the drivers guiding crop diversification of agropastoral systems in households in the ASB, and also to inform stakeholders (including policy-makers) through a portfolio of leverage points and processes needed to improve natural resources and livelihood resilience. Results of this study could be fed into the development of an aggregated system dynamics model to capture the livelihood contexts and key drivers of change and the development of the systems' scenarios.

As far as the current researchers understand, there have been no other studies conducted that have investigated the factors affecting rural households' decisions regarding crop diversification in the Central Asian region. As a demonstrative case, this study conducted research in the ASB (the Karauzyak district) because of its remote location, harsh environmental conditions, and relatively cold winters and hot summers that largely influence crop productivity and livestock.

## 2. Materials and Methods

### 2.1. Study Site

The Republic of Karakalpakstan (Figure 1) is northwest of Uzbekistan and embraces the vast dry lands in the lower reaches of the Amudarya River basin and the Aral Sea. Harsh environmental conditions in the study site, such as cold winters and hot summers, largely affect the productivity of crops and livestock, which is generally characterized as low. Reflecting on external conditions, the vulnerability of the livelihood system in Karakalpakstan is very high and the area is considered one of the regions with low incomes in Uzbekistan. Hence, crop and livestock production with ongoing land degradation and scarce irrigated water resources is a huge challenge for rural households in the ASB. To mitigate the negative impacts of the Aral Sea disaster, it is necessary to formulate optimal rural livelihood strategies via modelling current crop and livestock subsystems in selected sites [24]. This study selected the villages of Karabuga and Algabas to investigate, as these two villages represent the rural areas of the Karauzyak district in the Republic of Karakalpakstan.

One hundred households were randomly selected from a list of all households residing in the two villages. This constituted over 7% of the total households and that is assumed to be sufficient for such a reconnaissance study to obtain an overview of the villages and the district.

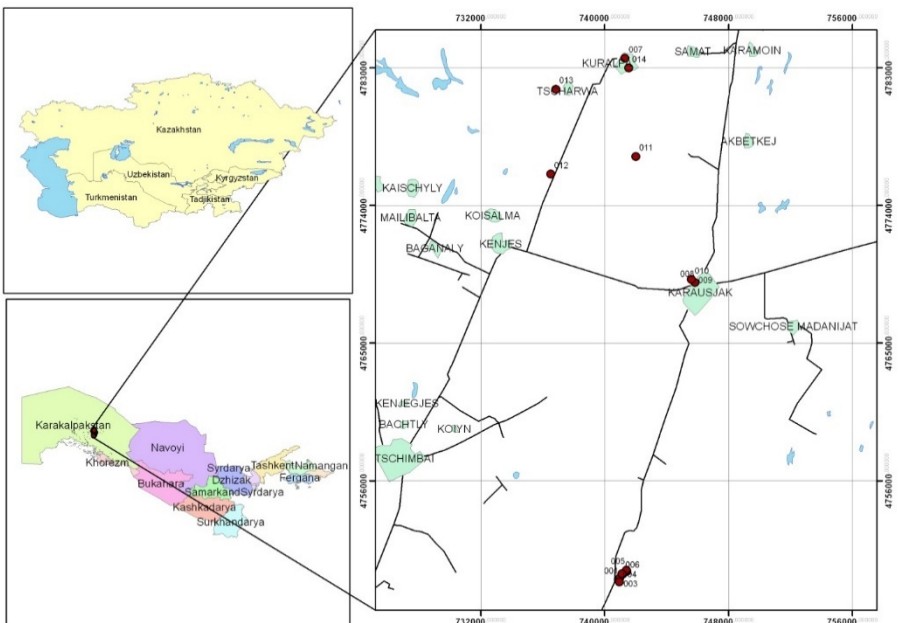

**Figure 1.** Location of the study area and the household surveys (Source: GIS lab of NGO "KRASS"). The study area is highlighted in the rectangular inset map. At the bottom left of the figure is the study area. At the right side of the figure, the villages are illustrated in green color, roads are expressed in lines and surveyed locations are demonstrated in maroon dots.

### 2.2. Household Farm Surveys and Analysis

The survey was prepared in correspondence with the SLF. A total of 100 households were randomly selected and interviewed. The consultants of the village citizen councils (females) helped to find interviewees, set contacts with local populations, and provided some local statistics. Since the interviews took place during the peak agricultural season, a combination of individual and group interviewing methodologies was applied.

Key informant interviews included the head of the local administration and his assistants, the head of the veterinary service, the heads of the village citizen councils of the selected areas (Karabuga and Algabas), and the consultants from the village citizen councils. The questionnaires covered household characteristics (e.g., demographics, education and professions), farmland inventories, land tenures, agricultural tool inventories, crop and livestock production, off-farm incomes, and remittances.

The main factors differentiating the households' agricultural livelihood systems (ALSs) were identified by using a Principal Component Analysis (PCA), and the results were used to further identify ALS types in cluster analyses (Figure 2). For the type-specific and overall drivers of production and livelihood diversifications, Shannon diversity indices were calculated per cluster type and the drivers were estimated using the Multiple Linear Regression (MLR) model.

### 2.3. Principle Component Analysis (PCA) and Subsequent Cluster Analysis (CA) for Identifying Types of Smallholder Agricultural Livelihood Systems (ALS)

We applied the empirical Typology-Based Approach described by Le and Dhehibi to analyze household data for defining ALS types [25]. To analyze type-specific behaviors of each ALS, livelihood types were determined in two steps. The first step included using a PCA to identify a few of the key variables representing different factors (i.e., being uncorrelated with each other) and to explain most of the variations among the existing datasets. Thus, the PCA helped reveal a limited number of principal components (PCs) (compared to all the original variables) representing the multivariate datasets; this provided a better focus on the subsequent households' cluster analyses and the households' type characterization. In this study, the variables used for the PCA were selected using the

SLF, which gave a broad explanation of the livelihoods of the poor and revealed the major factors influencing people's livelihoods [26]. The SLF distinguishes five livelihood assets (human, natural, financial, physical, and social capital), which this study selected as related factors from existing datasets. The variables representing the livelihood assets as entries for the PCA are shown in Table 1.

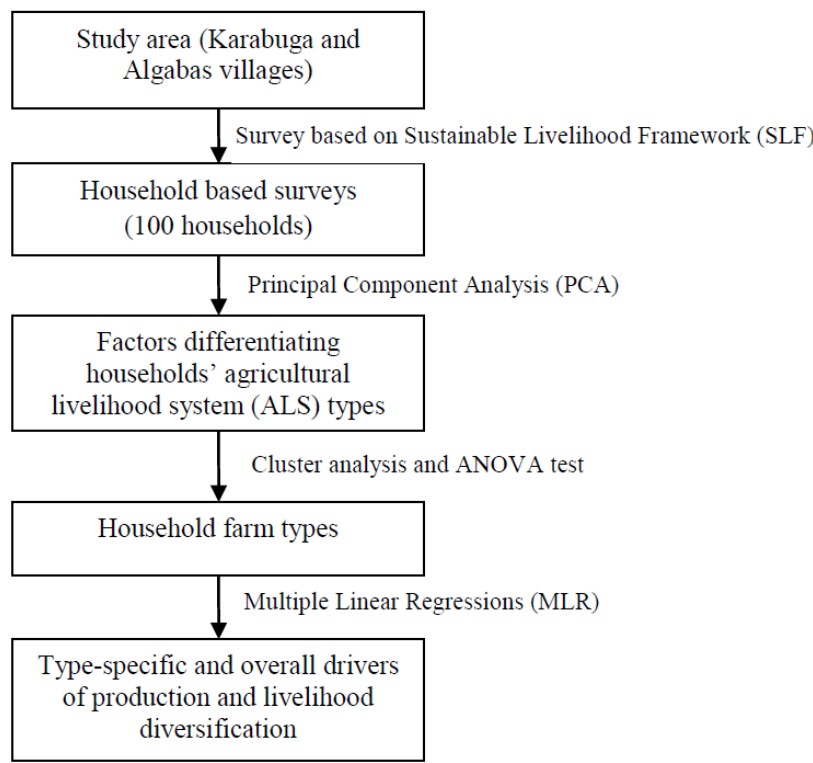

**Figure 2.** Flow chart of analytical steps.

**Table 1.** List of candidate variables for the PCA.

| Variables | Definition | Source * |
|---|---|---|
| | *Human asset* | |
| $H_{AGE}$ | Age of household head (years) | Direct |
| $H_{EXP}$ | Agriculture experience of household head (years) | Direct |
| $H_{SIZE}$ | Household family size (number of persons) | Direct |
| $H_{EDU}$ | Whether household has members with higher education (dummy) | Compound |
| $H_{LABOR}$ | Household labor (number of workers) | Direct |
| $H_{FLABOR}$ | Female household labor (number of workers) | Direct |
| $H_{PWORKERS}$ | Household potential workers (number of workers) | Compound |
| $H_{DEPRATIO}$ | Household dependency ratio (ratio between 0 and 1) | Compound |
| | *Financial asset* | |
| $H_{ONFARMINC}$ | Share of on-farm income in total income (%) | Direct |
| $H_{OFFFARMINC}$ | Share of off-farm income in total income (%) | Direct |
| $H_{NONAGROINC}$ | Share of non-agricultural income in total income (%) | Direct |
| $H_{LVSTUNIT}$ | Total livestock units [1] | Compound |
| $H_{CATTLE}$ | Share of cattle in total livestock (%) | Compound |
| $H_{RUMINANT}$ | Small ruminant share in total livestock (%) | Compound |
| $H_{POULTRY}$ | Poultry share in total livestock (%) | Compound |
| $H_{NONAGRO}$ | Household members with non-agricultural incomes (%) | Compound |
| $H_{AGRO}$ | Household members with agricultural incomes (%) | Compound |

**Table 1.** *Cont.*

| Variables | Definition | Source * |
|---|---|---|
| | *Natural asset* | |
| $H_{LAND}$ | Total landholding area (m$^2$) | Direct |
| $H_{LANDIRR}$ | Share of irrigated landholding (%) | Direct |
| $H_{LANDPC}$ | Land per capita area (m$^2$/person) | Compound |
| $H_{IRRIGATION}$ | Access to quality water for irrigation (clean water irrigation = 1, sewage irrigation = 0) | Compound |
| | *Physical asset* | |
| $H_{ASSET}$ | Housing asset index [2] | Compound |
| $H_{WELFARE}$ | Welfare score [3] | Compound |
| Social asset | | |
| $H_{SOCIAL}$ | Social capital (points) [4] | Compound |
| | *Production orientation* | |
| $H_{VEGETABLES}$ | Vegetable area (m$^2$) | Direct |
| $H_{WATERMELONS}$ | Watermelon area (m$^2$) | Direct |
| $H_{FODDER}$ | Fodder area (m$^2$) | Direct |
| $H_{FRUIT}$ | Fruit tree area (m$^2$) | Direct |
| | *Geographical variables* | |
| $H_{LVSTDIST}$ | Distance to livestock markets (km) | Direct |
| $H_{FOODDIST}$ | Distance to food markets (km) | Direct |

Note: * Direct = directly extracted from the survey, Compound = information calculated based on survey data. [1] To calculate total livestock units, livestock type was multiplied by respective coefficients and summed in total. Amount of mature sheep, rams, lambs, mature she-goats, he-goats, young animals were multiplied by 0.1, amount of turkeys by 0.03, amount of chickens and ducks by 0.014, and cattle, horses, and mules by 1. [2] Housing asset index is calculated by the following formula: HAI = (number of living rooms) / (total number of rooms) + (number of rooms with heating)2 + (number of rooms with electricity) / (total number of rooms). [3] Welfare score sums all a household's physical assets, where each type of asset is multiplied by a coefficient and discounted for its condition. Tractors and cars were multiplied by 10; water pump by 5; grain storage facility by 5; satellite antenna, refrigerator and furniture by 2; TV, radio, audio player, mobile phone and carpet by 1. For discounting, the item was divided by 2 if the condition was satisfactory and divided by 3 if the condition was bad.[4] Social capital score was calculated by adding the scores against the following criteria: 1 if (a) the household member was part of any public organization, or (b) the household could rely upon state subsidies in case of harvest loss at about 25%, or (c) the household's women (incl. single) could access services for extension agents, or (d) the household's women (incl. single) could access training and seminars on agriculture outside the communities; 2 if the household could rely upon state subsidies in case of harvest loss at about 50%; 3 if the household could rely upon state subsidies in case of harvest loss, then it received 1 point when loss was 75%; 4 if the household could rely upon state subsidies in case of harvest loss at 100%.

We conducted the PCA using the statistical package STATA (StataCorp., Stata Statistical Software: Release 11. College Station, TX, USA) The number of PCs were selected using the following two rules: first, PCs with eigenvalues equal to or larger than 1 should be retained, and second, PCs should represent sufficiently high shares of the total variations found in the entered multivariable datasets.

In the first step, the livelihood dimensions represented by each extracted PC were identified by the livelihood variables having high correlations with the PCs. These livelihood variables, rather than all the variables of the multivariate datasets, will also be used to characterize the household clusters/types identified in later steps.

The second step was a cluster analysis using the computed scores of the main PCs. A K-mean cluster analysis (K-CA) was selected for two reasons: (1) unlike hierarchical methods, K-CA methods avoid chaining and artificial boundary problems and work from the original input data rather than from a similarity matrix; and (2) the considered multivariate datasets are composed of several cases, making it difficult to interpret grouping results using a hierarchical cluster analysis. The optimal number of clusters was defined using the "knee curve" method as described by Le and Dhehibi [25]. With the "knee curve" method, we ran K-CAs with K = 2, 3, ... , 10 that used variables in Table 1, and calculated the sum of squared errors (SSE) for every K-CA run. The curve depicting the distribution of SSE versus the running number of clusters (K) shows a "knee" point at a particular K* value

suggesting an optimal cluster number. Increasing the cluster number further from this K* will not effectively increase the average clumsiness of each cluster [25].

### 2.4. Agricultural Production and Livelihood Diversity Indices

The study measured the diversities of the households' income sources, crop and livestock productions and their compositions to see whether the ALS types were significantly different from each other in terms of diversification strategies. This study used the Shannon diversity index [27] which uses the following form:

$$H = - \sum_{i=1}^{S} p_i ln p_i \qquad (1)$$

where $H$ is the diversity index for a set, including $S$ component types, and $p_i$ is the abundant coefficient of component type $i$ ($i$ = 1, 2, 3, ... $S$).

For each household, the Shannon indices of the three following types of diversities were calculated:

- The diversity of household income sources ($H_{INCOME}$): The number of component types are the three main types of income sources that are on-farm, off-farm agricultural, and off-farm non-agricultural. The abundant coefficient is the share (in %) of each income source compared to the total household income (100%);
- The diversity of crop production ($H_{CROP}$): The number of component types is the four main types of crops, which are vegetable, watermelon, fodder, and fruit. The abundant coefficient is the share of the area (in %) of each main crop type compared to the total household cropping crop area (100%);
- The diversity of livestock production ($H_{LIVESTOCK}$): The number of component types is the three main types of livestock in the region, which are cattle, small ruminants, and poultry. The abundant coefficients are the share of livestock units (in %) of each livestock type compared to the total livestock units owned by the household (100%).

### 2.5. Inferential Statistics for Identifying Drivers of Diversification

The MLR model was used to identify the driving factors of crop diversification. The model was used for each type of ALS and the total sampled populations. This revealed the group-specific determinants of each household's livelihood diversifications compared to treating all the households as a group. Due to the high number of explaining variables in the model, the Variance Inflation Factor (VIF) test was run to check for potential multicollinearity in the regression model. In the cases when the VIF test indicated a high correlation between the explaining variables, the model was improved by excluding the highly correlating and the comparably less relevant variables.

Household diversification indices that were found to be statistically different between the ALS types were entered in the regression model as the response variables (Table 2).

**Table 2.** List of response and explaining variables in multiple linear regression (MLR) analyses.

| Variables | Definition | Hypothesized Effect * |
|---|---|---|
| | **Response variables** | |
| $H_{CROP}$ | Shannon diversity index of crop production | |
| $H_{LIVESTOCK}$ | Shannon diversity index of livestock production | |
| $H_{INCOME}$ | Shannon diversity index of income sources | |
| | **Explaining variables** | |
| Human asset | | |
| $H_{EXP}$ | Agricultural experiences of household heads (years) | + |
| $H_{PWORKERS}$ | Household's potential workers (number of workers) | + |
| $H_{EDU}$ | Whether households have members with higher education (yes = 1, no = 0) | + |
| $H_{DEPRATIO}$ | Household's dependency ratio (ration between 0 and 1) | +/− |

**Table 2.** *Cont.*

| Variables | Definition | Hypothesized Effect * |
|---|---|---|
| *Financial asset* | | |
| $H_{CATTLE}$ | Share of cattle in total livestock (%) | − |
| $H_{RUMINANT}$ | Small ruminant share in total livestock (%) | − |
| $H_{ONFARMINC}$ | Share of on-farm income in total income (%) | + |
| *Natural asset* | | |
| $H_{LAND\ (ha)}$ | Total area of land (hectare) | + |
| $H_{IRRIGATION}$ | Access to quality water for irrigation (clean water irrigation = 1, sewage irrigation = 0) | + |
| *Social asset* | | |
| $H_{SOCIAL}$ | Social capital (points) | + |
| *Geographic* | | |
| $H_{FOODDIST}$ | Distance to livestock markets (km) | +/− |
| $H_{LVSTDIST}$ | Distance to food markets (km) | +/− |

Note: * + and − indicate positive and negative effects, respectively. +/− indicates unclear/no effect.

The explanatory variables (Table 2) that were entered in the multiple regression model were identified by plausible theories and regional settings. Several relevant variables, which should also have impacted the response variables, were dropped from the model to avoid multicollinearity. The entered variables were the following:

*Human Assets:* The agricultural experiences of the household heads ($H_{EXP}$) gave a better understanding of the cultivation of different crops and, hence, is hypothesized to positively influence the households to diversify their crop production. Households with more potential workers ($H_{PWORKERS}$) were able to cultivate any type of crop (requiring a larger or smaller labor force) without being restricted to crops requiring a smaller labor force, which is hypothesized to have a positive effect on crop diversification. Households with highly educated members ($H_{EDU}$) should have rational approaches toward agricultural activity, hence diversifying its crop production. A high dependency ratio ($H_{DEPRATIO}$) in the household might restrict household members to work more on agricultural plots and concentrate only on crops requiring less labor or crops that have high investment returns. On the other hand, a high dependency ratio might push households to cultivate more types of crops to be more self-sufficient. The true effect of the dependency ratio can be seen in this study's results (Table 2);

*Financial Assets:* High shares of cattle ($H_{CATTLE}$) and small ruminants ($H_{RUMINANT}$) within the total amount of livestock usually drive the household to cultivate mainly fodder and less diverse crops in order to feed the livestock. High shares of on-farm income (HONFARMINC) in the household's total income might indicate that the household is mainly involved in farm production and would have diverse agricultural activity;

*Natural Assets:* Holding larger plots of land ($H_{LAND}$ (ha)) gives the households more potential to cultivate other types of crops to reach the optimum levels of cultivation area per crop. Access to quality water for irrigation ($H_{IRRIGATION}$) provides an opportunity to cultivate crops requiring proper irrigation. Hence, households with access to clean water for irrigation have more opportunities to cultivate the preferred types of crops compared to households that do not have the access and are restricted to only the crops requiring lower quality irrigation water. It is expected that households with access to good irrigation would have more diverse crops;

*Social Assets*: Having a higher social capital ($H_{SOCIAL}$) increases accessibility to necessary institutions, inflow of information (extension), and mutual cooperation that would eventually lead to better farming and the production of more diverse crops. The social capital of each household was evaluated by using criteria such as leadership, membership in public organizations, access to state subsidies, access of the women to extension services and seminars and training on agriculture outside the community;

*Physical Assets:* Physical access of household to a food market is approximated by the distance from household farm to the nearest market ($H_{FOODDIST}$). Increased distance to food markets ($H_{FOODDIST}$) makes it difficult for households to commute to the markets to

sell or buy food. On one hand, far distance might push households to diversify their crops to be more self-sufficient in food. On the other hand, it might lead to cultivating crops that are convenient to transport.

The MLR model's equation is as follows:

$$\begin{aligned} H = \beta_0 + \beta_1 H_{EXP} \quad &+ \beta_2 H_{PWORKERS} + \beta_3 H_{EDU} + \beta_4 H_{DEPRATIO} + \beta_5 H_{CATTLE} \\ &+ \beta_6 H_{RUMINANT} + \beta_7 H_{ONFARMINC} + \beta_8 H_{LAND(HA)} \\ &+ \beta_9 H_{IRRIGATION} + \beta_{10} H_{SOCIAL} + \beta_{11} H_{FOODDIST} + \beta_{12} H_{LVSTDIST} \\ &+ e \end{aligned}$$

where $H$ is the Shannon diversity index as the response variable, $H_{EXP}$, $H_{PWORKERS}$, ..., $H_{LVSTDIST}$ are the explaining livelihood variables (see Table 2), $\beta_0$, $\beta_1$, ..., $\beta_{12}$ are the parameters to be estimated by the multiple linear regression, and $e$ is an error term.

## 3. Results

### 3.1. Key Variables Representing Smallholders' Agricultural Livelihoods

The PCA revealed the main factors affecting smallholder systems. In the PCA, 11 PCs explaining 74.8% of the total variances were selected (Table 3). To determine the PC loadings, orthogonal rotation was applied. In Table 3, the PCs were labeled after the variables with the highest loadings within each component (with bolt numbers). The most apparent factors among the surveyed households that had at least 8% of the initial variances were PC-1, PC-2, and PC-3. PC-1 had the highest initial variance (9.7%) and highly correlated with the households' labor amount (loading = 0.56). This represented the human assets of the households, hence being labeled Labor PC. Next, PC-2 had 9.3% of the initial variance and was highly correlated with household members who had non-agricultural income (loading = 0.54). This represented the financial assets of the households and was labeled as Non-agricultural Members PC. PC-3 had 8.5% of the initial variance and highly correlated with land per capita (loading = 0.58), which represented the physical assets of the households. Subsequently, it was labeled as Land per capita PC. The remaining PCs, from PC-4 to PC-11, each had initial variances ranging from 4% to 8% and were labeled as factors of higher education and housing asset (PC-4), the head of the households' ages (PC-5), cattle shares (PC-6), on-farm incomes (PC-7), distance from food markets (PC-8), distance from livestock markets (PC-9), household dependency ratios (PC-10), and off-farm incomes (PC-11), respectively, in accordance with their highly correlated variables within the components.

**Table 3.** Key components and variables representing agricultural livelihoods of smallholders in the Karauzyak district.

| | PC-1: Labor | PC-2: Non-Agr Members | PC-3: Land per Capita | PC-4: High Edu and Housing Asset | PC-5: Hh Head Age | PC-6: Cattle Share | PC-7: On-Farm Income | PC-8: Food Market Distance | PC-9: Livestock Market Distance | PC-10: Hh Dep. Ratio | PC-11: Off-Farm Income |
|---|---|---|---|---|---|---|---|---|---|---|---|
| | *9.7%* | *9.3%* | *8.5%* | *7.9%* | *6.9%* | *6.2%* | *6.1%* | *5.6%* | *5.4%* | *4.6%* | *4.6%* |
| *Human asset* | | | | | | | | | | | |
| $H_{AGE}$ | −0.05 | 0.03 | −0.03 | −0.01 | 0.67 | −0.03 | −0.02 | −0.01 | 0.01 | −0.01 | 0.02 |
| $H_{EXP}$ | 0.05 | 0.02 | 0.05 | −0.04 | 0.57 | 0.09 | 0.00 | −0.01 | 0.01 | −0.03 | 0.01 |
| $H_{SIZE}$ | 0.32 | −0.11 | −0.10 | 0.19 | 0.17 | −0.05 | 0.01 | −0.06 | −0.03 | 0.29 | −0.15 |
| $H_{EDU}$ | 0.03 | 0.12 | −0.09 | 0.43 | −0.05 | −0.32 | −0.12 | −0.09 | −0.10 | −0.15 | −0.01 |
| $H_{LABOR}$ | 0.56 | 0.07 | 0.01 | −0.05 | −0.03 | 0.07 | 0.03 | −0.06 | −0.06 | 0.05 | 0.05 |
| $H_{FLABOR}$ | 0.51 | 0.01 | −0.01 | −0.05 | −0.10 | 0.18 | −0.12 | −0.01 | 0.05 | 0.04 | −0.02 |
| $H_{PWORKERS}$ | 0.44 | −0.07 | 0.00 | 0.07 | 0.14 | −0.18 | 0.07 | 0.06 | 0.04 | −0.17 | −0.02 |
| $H_{DEPRATIO}$ | −0.27 | 0.06 | −0.12 | 0.09 | −0.03 | 0.23 | −0.05 | −0.16 | −0.03 | 0.55 | −0.08 |

**Table 3.** *Cont.*

| | PC-1: Labor | PC-2: Non-Agr Members | PC-3: Land per Capita | PC-4: High Edu and Housing Asset | PC-5: Hh Head Age | PC-6: Cattle Share | PC-7: On-Farm Income | PC-8: Food Market Distance | PC-9: Livestock Market Distance | PC-10: Hh Dep. Ratio | PC-11: Off-Farm Income |
|---|---|---|---|---|---|---|---|---|---|---|---|
| | *9.7%* | *9.3%* | *8.5%* | *7.9%* | *6.9%* | *6.2%* | *6.1%* | *5.6%* | *5.4%* | *4.6%* | *4.6%* |
| *Financial asset* | | | | | | | | | | | |
| $H_{ONFARMINC}$ | 0.00 | −0.05 | −0.04 | −0.03 | 0.00 | 0.00 | 0.68 | 0.00 | 0.01 | 0.02 | −0.20 |
| $H_{OFFFARMINC}$ | −0.01 | −0.20 | −0.07 | 0.04 | 0.03 | 0.08 | −0.16 | 0.05 | −0.02 | 0.03 | 0.62 |
| $H_{NONAGROINC}$ | 0.01 | 0.24 | 0.04 | −0.07 | 0.05 | −0.08 | −0.44 | 0.05 | −0.07 | 0.06 | −0.22 |
| $H_{LVSTUNIT}$ | −0.03 | 0.02 | 0.04 | 0.40 | 0.13 | 0.10 | 0.11 | 0.07 | 0.05 | 0.11 | 0.05 |
| $H_{CATTLE}$ | 0.06 | −0.02 | 0.07 | 0.08 | 0.01 | 0.57 | −0.02 | 0.13 | 0.04 | −0.06 | −0.10 |
| $H_{RUMINANT}$ | −0.05 | −0.09 | −0.01 | 0.41 | −0.15 | 0.09 | 0.10 | −0.02 | 0.09 | −0.16 | 0.03 |
| $H_{POULTRY}$ | −0.05 | −0.10 | 0.03 | −0.03 | −0.06 | −0.54 | −0.06 | 0.12 | 0.12 | −0.01 | −0.16 |
| $H_{NONAGRO}$ | −0.01 | 0.54 | −0.01 | −0.01 | 0.03 | 0.02 | −0.06 | 0.00 | 0.05 | 0.00 | −0.02 |
| $H_{AGRO}$ | 0.01 | −0.54 | 0.01 | 0.01 | −0.03 | −0.02 | 0.06 | 0.00 | −0.05 | 0.00 | 0.02 |
| *Natural asset* | | | | | | | | | | | |
| $H_{LAND}$ | 0.03 | −0.07 | 0.57 | 0.07 | 0.08 | −0.01 | −0.05 | 0.00 | −0.05 | −0.02 | −0.06 |
| $H_{LANDIRR}$ | 0.02 | 0.17 | 0.08 | −0.01 | −0.02 | −0.11 | 0.17 | 0.06 | 0.54 | 0.08 | −0.08 |
| $H_{LANDPC}$ | −0.07 | 0.00 | 0.58 | −0.06 | 0.01 | 0.02 | −0.06 | 0.01 | −0.05 | −0.14 | 0.04 |
| $H_{IRRIGATION}$ | −0.03 | 0.07 | 0.01 | 0.06 | 0.00 | 0.06 | 0.02 | 0.54 | 0.07 | −0.03 | 0.27 |
| *Physical asset* | | | | | | | | | | | |
| $H_{ASSET}$ | 0.04 | 0.00 | 0.04 | 0.42 | −0.07 | 0.02 | −0.24 | 0.10 | 0.00 | 0.08 | −0.04 |
| $H_{WELFARE}$ | 0.08 | 0.29 | −0.01 | 0.32 | −0.11 | 0.00 | 0.17 | −0.08 | −0.14 | −0.10 | 0.14 |
| *Social capital* | | | | | | | | | | | |
| $H_{SOCIAL}$ | −0.04 | 0.21 | 0.11 | 0.07 | 0.15 | −0.21 | 0.27 | 0.04 | −0.07 | 0.15 | 0.43 |
| *Production orientation* | | | | | | | | | | | |
| $H_{VEGETABLES}$ | −0.02 | 0.07 | −0.27 | −0.02 | 0.04 | 0.10 | −0.01 | 0.17 | 0.49 | −0.16 | −0.04 |
| $H_{WATERMELON}$ | −0.09 | −0.14 | 0.03 | 0.31 | 0.18 | 0.00 | 0.00 | 0.06 | −0.04 | 0.14 | −0.32 |
| $H_{FODDER}$ | 0.05 | 0.08 | 0.43 | 0.01 | −0.15 | 0.08 | 0.11 | 0.00 | 0.22 | 0.18 | −0.03 |
| $H_{FRUIT}$ | −0.15 | 0.02 | −0.02 | 0.11 | 0.08 | 0.18 | 0.00 | −0.15 | −0.02 | −0.61 | −0.14 |
| *Geographical variables* | | | | | | | | | | | |
| $H_{LVSTDIST}$ | −0.01 | −0.19 | 0.07 | 0.11 | 0.05 | −0.05 | −0.21 | −0.33 | 0.57 | 0.03 | 0.17 |
| $H_{FOODDIST}$ | −0.01 | −0.09 | 0.00 | 0.03 | −0.01 | −0.03 | −0.07 | 0.66 | −0.02 | 0.04 | −0.13 |

## 3.2. Main Agricultural Livelihood Types

Based on the PCA and the k-means cluster analysis, this study identified three types of ALSs in the study site. As per the descriptive statistics provided in Table 4, the following types of ALSs can be characterized:

**Table 4.** Descriptive statistics of key agricultural livelihood system (ALS) variables of three identified smallholder types with ANOVA tests.

| Variable | ALS Type 1: 26 Observations | | ALS Type 2: 31 Observations | | ALS Type 3: 43 Observations | |
|---|---|---|---|---|---|---|
| | **Mean** | **CI$_{0.05}$** | **Mean** | **CI$_{0.05}$** | **Mean** | **CI$_{0.05}$** |
| *Human asset* | | | | | | |
| $H_{AGE}$ | 55 [a] | ±4.499 | 54 [a] | ±3.142 | 43 [b] | ±2.398 |
| $H_{EXP}$ | 29 [a] | ±4.010 | 29 [a] | ±4.045 | 16 [b] | ±2.854 |
| $H_{SIZE}$ | 6 [a] | ±0.542 | 7 [b] | ±0.613 | 5 [c] | ±0.369 |
| $H_{EDU}$ | 0.54 [a] | ±0.205 | 0.52 [a] | ±0.186 | 0.23 [b] | ±0.131 |
| $H_{LABOR}$ | 3 [a] | ±0.422 | 5 [b] | ±0.541 | 2 [c] | ±0.254 |
| $H_{FLABOR}$ | 1 [a] | ±0.200 | 2 [b] | ±0.315 | 1 [a] | ±0.090 |
| $H_{PWORKERS}$ | 3 [a] | ±0.548 | 5.19 [b] | ±0.458 | 2 [c] | ±0.239 |
| $H_{DEPRATIO}$ | 0.86 [a] | ±0.277 | 0.31 [b] | ±0.116 | 0.80 [a] | ±0.181 |

**Table 4.** *Cont.*

| Variable | ALS Type 1: 26 Observations | | ALS Type 2: 31 Observations | | ALS Type 3: 43 Observations | |
|---|---|---|---|---|---|---|
| | **Mean** | **CI$_{0.05}$** | **Mean** | **CI$_{0.05}$** | **Mean** | **CI$_{0.05}$** |
| *Financial asset* | | | | | | |
| $H_{ONFARMINC}$ | 11.27 | ±7.485 | 12.52 | ±6.495 | 17.09 | ±8.841 |
| $H_{OFFFARMINC}$ | 2.69 | ±4.206 | 5.00 | ±5.208 | 7.33 | ±6.735 |
| $H_{NONAGROINC}$ | 86.04 | ±8.441 | 79.26 | ±9.121 | 71.63 | ±11.130 |
| $H_{LVSTUNIT}$ | 9 [a] | ±2.926 | 3 [b] | ±1.627 | 2 [b] | ±1.061 |
| $H_{CATTLE}$ | 0.94 [a] | ±0.015 | 0.72 [b] | ±0.147 | 0.60 [b] | ±0.143 |
| $H_{RUMINANT}$ | 0.03 | ±0.012 | 0.02 | ±0.018 | 0.02 | ±0.014 |
| $H_{POULTRY}$ | 0.03 [ab] | ±0.014 | 0.10 [bc] | ±0.090 | 0.20 [c] | ±0.118 |
| $H_{NONAGRO}$ | 0.39 [a] | ±0.073 | 0.30 [a] | ±0.065 | 0.21 [b] | ±0.042 |
| $H_{AGRO}$ | 0.61 [a] | ±0.073 | 0.70 [a] | ±0.065 | 0.79 [b] | ±0.042 |
| *Natural asset* | | | | | | |
| $H_{LAND}$ | 2938 [a] | ±423.326 | 1238 [b] | ±356.595 | 1527 [b] | ±257.69 |
| $H_{LANDIRR}$ | 0.63 | ±0.100 | 0.59 | ±0.096 | 0.50 | ±0.095 |
| $H_{LANDPC}$ | 555 [a] | ±119.214 | 189 [b] | ±54.029 | 351 [c] | ±65.039 |
| $H_{IRRIGATION}$ | 0.58 | ±0.203 | 0.35 | ±0.178 | 0.47 | ±0.155 |
| *Physical asset* | | | | | | |
| $H_{ASSET}$ | 16.72 [a] | ±5.236 | 8.96 [b] | ±4.523 | 4.49 [b] | ±1.939 |
| $H_{WELFARE}$ | 12.56 [a] | ±3.491 | 7.93 [b] | ±1.631 | 5.84 [b] | ±1.345 |
| *Social asset* | | | | | | |
| $H_{SOCIAL}$ | 3.19 [a] | ±0.636 | 2.03 [b] | ±0.291 | 2.12 [b] | ±0.278 |
| *Production orientation* | | | | | | |
| $H_{VEGETABLES}$ | 0.15 [a] | ±0.058 | 0.40 [b] | ±0.123 | 0.17 [a] | ±0.075 |
| $H_{WATERMELONS}$ | 0.07 [a] | ±0.042 | 0.02 [b] | ±0.016 | 0.01 [b] | ±0.011 |
| $H_{FODDER}$ | 0.29 [a] | ±0.109 | 0.03 [b] | ±0.033 | 0.09 [b] | ±0.056 |
| $H_{FRUIT}$ | 0.03 | ±0.024 | 0.02 | ±0.031 | 0.01 | ±0.010 |
| *Geographical variables* | | | | | | |
| $H_{LVSTDIST}$ | 15.17 | ±3.188 | 18.95 | ±3.154 | 14.57 | ±3.059 |
| $H_{FOODDIST}$ | 18.19 | ±3.018 | 14.54 | ±3.943 | 17.25 | ±2.747 |

Note: Among ALS types (i.e., in each row), mean values with the same alphabet letter indicate not significantly different at the confidence level of 95%.

ALS type 1: Land and livestock per capita rich (Figure 3). Around a quarter (26%) of the households belonged to this livelihood type. The ALS type 1 households had relatively large landholdings with average land areas of 2938 m$^2$, owned large amounts of livestock units, most of which constituted cattle, and had high shares of off-farm income. In addition to these major differentiating factors, these households were also rich in housing assets, had higher social capital, and had better welfare. Watermelons and fodder crops were more commonly cultivated by these types of households. The ages, experience, and education of the household heads were high in these types of households. They were similar to ALS type 2 households and much higher than ALS type 3 households. However, the available labor force was not proportionate to the landholding size, and subsequently the dependency ratio in this ALS type was high;

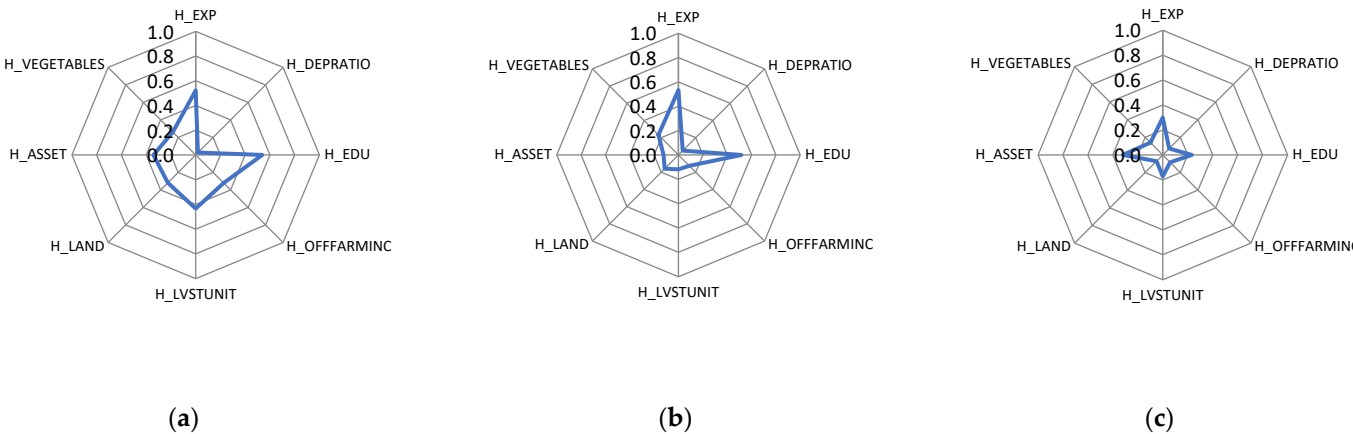

**Figure 3.** Spider diagram of indicators (standardized score) for the three identified agricultural livelihood system (ALS) types. Note: The variables included in the figure were selected from Table 4 with the following criteria: representing 5 livelihood assets and statistically significant differences among ALS types. (**a**) Livelihood type 1. (**b**) Livelihood type 2. (**c**) Livelihood type 3.

ALS type 2: Relatively labor rich, poorer land per capita, and lower dependency ratio (Figure 4). Around one third (31%) of the sample size consisted of these types of households. In these types households had large family sizes, with 7 members on average. This translated to a larger work force, with 5 workers on average. Land possession per capita was smaller than in other household types (189 m2 per family member). Consequently, livestock units in these households were low. Despite having a large family, the ages and experiences of the household heads were high. More of the household heads had higher education as well, which translated into a much lower dependency ratio. Vegetables were the main crop types cultivated by these households. Household assets were the lowest among these ALS types. The shares from off-farm incomes were much lower than in ALS type 1 households but were similar to ALS type 3 households;

ALS type 3: Relatively young, fewer members with non-agricultural income, and less labor (Figure 3c). The majority of the households (43%) belonged to this group. These households had relatively young family heads, therefore consisting of younger families with less experience and education. Most of the household members only had agricultural incomes, and a few of the members had non-agricultural income. Considering their small family sizes, these types of households had less labor and higher dependency ratios. These households had little land, which was occupied mainly by vegetables and larger numbers of poultry than in other ALS types.

The differences among these households' economic viabilities can be visualized through spider charts, where the size of the encircled area reflects the natural, physical, financial, and social endowments of the identified ALS types. The encircled area size of ALS type 1 households was wide and large, ALS type 2 was intermediate, and ALS type 3 was constricted.

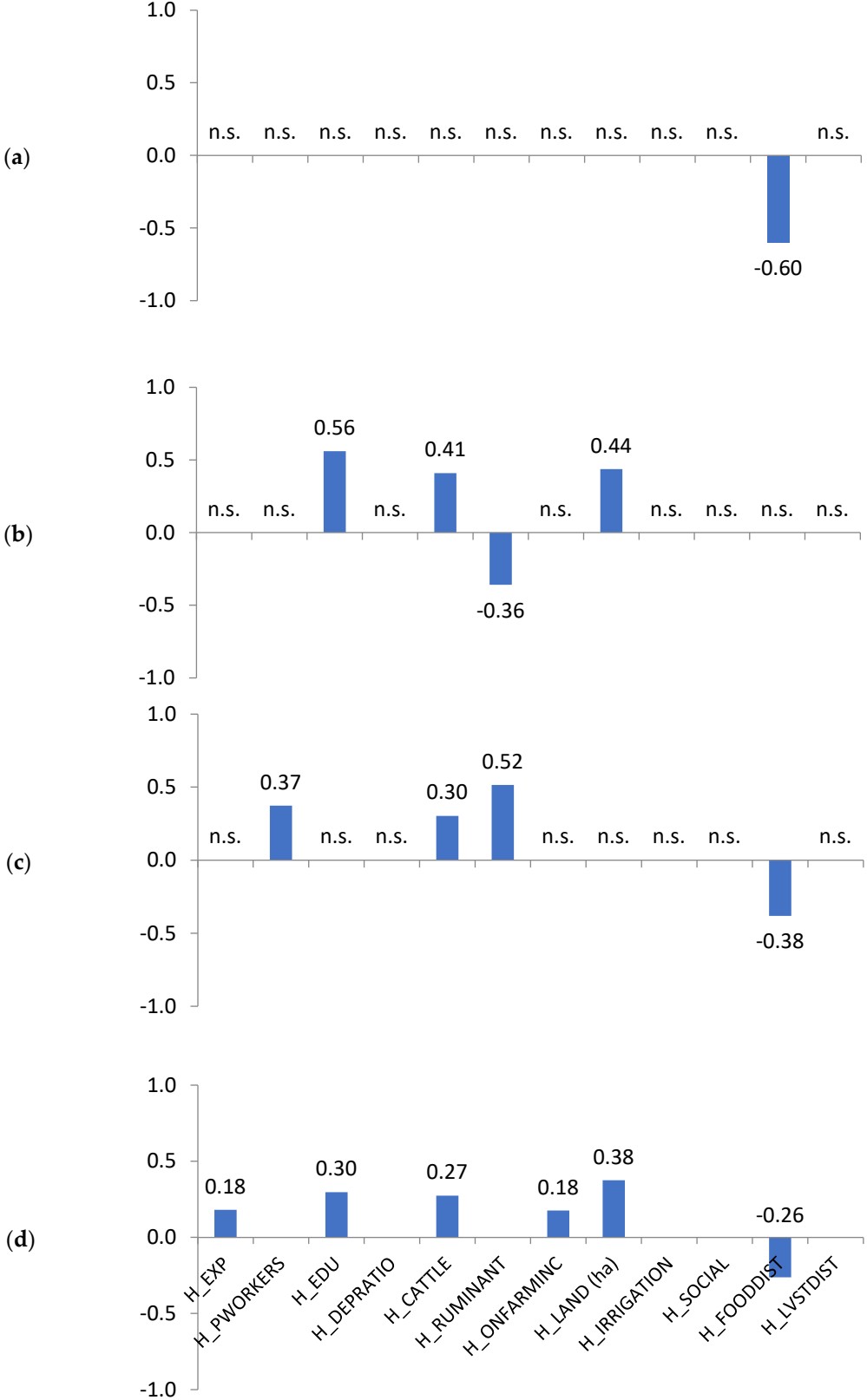

**Figure 4.** Bar charts showing significant drivers, their affecting directions, and magnitudes of production diversification by agricultural livelihood types. (**a**) ALS type 1. (**b**) ALS type 2. (**c**) ALS type 3. (**d**) Total sampled population. Note: n.s. stands for "not significant". Standardized beta coefficients are given in the figure.

### 3.3. Type-Specific and Overall Drivers of Production and Livelihood Diversification

The crop diversification index ($H_{CROP}$) was entered into the model as the response variable that was found to be significantly different from the identified ALS types (Table 5). The income diversification index ($H_{INCOME}$) and the livestock diversification index ($H_{LIVESTOCK}$) were found to be insignificantly different from the ALS types, and the regression model estimations for these response variables showed very poor model performance.

**Table 5.** Agricultural livelihood systems (ALS) type-specific Shannon diversity indices with ANOVA tests.

| | Agricultural Livelihood Type 1: 26 Observations | | Agricultural Livelihood Type 2: 31 Observations | | Agricultural Livelihood Type 3: 43 Observations | | ANOVA |
|---|---|---|---|---|---|---|---|
| **Variable** | **Mean** | **CI$_{0.05}$** | **Mean** | **CI$_{0.05}$** | **Mean** | **CI$_{0.05}$** | **Prob > F** |
| $H_{INCOME}$ | 0.26 | ±0.119 | 0.32 | ±0.108 | 0.23 | ±0.092 | 0.3978 |
| $H_{CROP}$ | 0.61 [a] | ±0.123 | 0.33 [b] | ±0.100 | 0.28 [b] | ±0.088 | 0.0000 |
| $H_{LIVESTOCK}$ | 0.21 [a] | ±0.052 | 0.17 [ab] | ±0.070 | 0.11 [b] | ±0.051 | 0.0513 |

Note: Among ALS types (i.e., in each row), mean values with the same alphabet letter indicate not significantly different at the confidence level of 95%.

The results from the regression analysis on the determinants of diversifying crop production are provided in Table 6. The model is statistically significant for the total and individual ALS types, where $R^2$ ranged from 0.50–0.71, indicating that the variables explained more than half of the variations in the model. Only significant drivers per group were reported in the table with their statistical significance levels.

**Table 6.** Determinants of diversification of crop production by ALS types.

| | **Total** | **ALS Type 1** | **ALS Type 2** | **ALS Type 3** |
|---|---|---|---|---|
| *Human asset* | | | | |
| $H_{EXP}$ | 0.005 * (0.002) | n.s. | n.s. | n.s. |
| $H_{PWORKERS}$ | n.s. | n.s. | n.s. | 0.137 * (0.052) |
| $H_{EDU}$ | 0.191 ** (0.052) | n.s. | 0.301 ** (0.087) | n.s. |
| $H_{DEPRATIO}$ | n.s. | n.s. | n.s. | n.s. |
| *Financial asset* | | | | |
| $H_{CATTLE}$ | 0.216 ** (0.067) | n.s. | 0.279 * (0.118) | 0.186 * (0.074) |
| $H_{RUMINANT}$ | n.s. | n.s. | −1.967 * (0.914) | 3.199 ** (0.738) |
| $H_{ONFARMINC}$ | 0.002 * (0.001) | n.s. | n.s. | n.s. |
| *Natural asset* | | | | |
| $H_{LAND\ (ha)}$ | 1.029 ** (0.230) | n.s. | 1.228 * (0.503) | n.s. |
| $H_{IRRIGATION}$ | n.s. | n.s. | n.s. | n.s. |
| *Social asset* | | | | |
| $H_{SOCIAL}$ | n.s. | n.s. | n.s. | n.s. |
| *Geographic* | | | | |
| $H_{FOODDIST}$ | −0.009 ** (0.003) | −0.024 * (0.008) | n.s. | −0.012 ** (0.003) |
| $H_{LVSTDIST}$ | n.s. | n.s. | n.s. | n.s. |
| constant | n.s. | n.s. | n.s. | n.s. |
| Observations (n) | 100 | 26 | 31 | 43 |
| Prob > F | 0.000 | 0.097 | 0.042 | 0.000 |
| R-squared | 0.507 | 0.662 | 0.621 | 0.717 |
| Adj R-squared | 0.439 | 0.350 | 0.368 | 0.604 |

Note: n.s. indicates non-significant (95%); * and ** indicate p-value ≤ 0.1 and 0.05 respectively.

The results for the total ALS (Table 4) demonstrate that the household heads' agricultural experiences and high levels of education positively impacted the households' crop diversifications. Contrary to this study's hypothesis, another factor that drove the households to diversify their crop productions was the shares of cattle in the total amounts

of livestock. Shares of on-farm incomes and landholding sizes positively impacted crop diversification. Lastly, distance to food markets negatively impacted crop diversification.

In ALS type 1 households, land- and livestock-rich households only had one significant negative effect, which was their distances from food markets. The regression results for ALS type 2 households revealed that households with educated members, large cattle shares and landholding sizes had positive impacts. Drivers that had negative impacts in ALS type 2 were shares of small ruminants. The results for ALS type 3 households indicated that the number of the households' potential workers, shares of small ruminants, and shares of cattle had positive impacts, and the households' distances to food markets had negative impacts.

Access to quality irrigation water, social capital and distances to livestock markets showed no statistically significant impact on any of the ALS types or on the total ALS. Interestingly, shares of small ruminants had different impacts on different ALS types. It had negative impacts on ALS type 2 households and positive impacts on ALS type 3 households. Overall, households with educated members, shares of cattle and ruminants, landholding size and distance to food markets were the common drivers that influenced the decision of the households to diversify their crops.

## 4. Discussion

### 4.1. Validity of the Identified Typology of Smallholder ALSs

The validity of smallholder, farm-household grouping in the presented study is supported by the similarities between the presented results with the findings of other studies in neighboring areas. In Khorezm province, ref. [22] identified three groups of households: two small groups of households that were relatively well- or poorly endowed, and a moderately endowed group with majority of households. Proportionately, the ALS types identified in our study had similar distribution with the well-endowed, ALS type 1 households that were slightly less in size, and the less-endowed ALS type 3 households that were slightly larger in size compared to findings by [22]. The moderately endowed group of households in the Khorezm province had at least one cow and possibly a vehicle, whereas in Karakalpakstan, a vehicle was more attributable to well-endowed, ALS type 1 households. On the other hand, the less-endowed household group in Khorezm was mainly female-headed with several children and only 1 or 2 working adults; their incomes were either irregular or mostly in-kind, allowing very little opportunity to diversify agricultural activities [22]. Comparable ALS type 3 households in Karakalpakstan had less assets compared to ALS type 1 and 2 households, but at least 3 out of 5 households had cattle in terms of livestock units, which was considered second only to land as an important form of physical capital for rural families worldwide [28].

### 4.2. Role of Non-Agricultural Activities

Interestingly, even though off-farm income was noted fewer times and members with agricultural incomes were twice as frequent as members with non-agricultural incomes, in all the ALS type households the major shares of income came from non-agricultural activities (Table 4). This fact might indicate two points: (1) agricultural incomes are substantially low; and (2) most of the agricultural products produced by the households are not marketed for sale but are produced for household consumption.

High share of non-agricultural income in households is noteworthy, despite agricultural production in rural settings being one of the key livelihood strategies for households; the evidence from this study is comparable to observations by [17] that stated the wages from agricultural activities in the neighboring province of Khorezm were rarely reported as major sources of income. Studies indicated that households relying on on-farm activities usually generated more in-kind income than cash income [22] and produced crops and livestock for subsistence purposes [29]. Furthermore, the households did not view in-kind incomes as noteworthy livelihood options, meaning they could be under-reported [17]. Estimating the households' in-kind incomes within local contexts is a complex task.

### 4.3. Main Livelihood Drivers for Diversification Strategies of Smallholders in the Study Region

Landholding size seems to be one of the strongest drivers overall for ALSs, particularly for ALS type 2 households, which had the smallest amount of land per capita. There are various interventions that were tested in the region and could help these households develop crop diversification strategies. Among those interventions was short-rotation plantation forestry that could help mitigate the repercussions of water shortages on rural livelihoods while sustaining energy needs, income, and food security [29]. Authors indicated that integration of such activities into clean development mechanisms could draw rewards from carbon sequestration, and increase profits following the harvest of tree plantations by transmitting the funds to rural households through existing wage-labor payment arrangements. Given the rural context and remoteness of the study areas, the issue of access to fuelwood is crucial. Fuelwood collection is often left to the women and children, which diverts them from obtaining adequate education. Education could later be a crucial driver for livelihood diversification. Analyses indicated that households with higher education had more diverse crops.

Not surprisingly, livestock had a significant impact on crop diversification, indicating the need for feed crops. However, fruit trees in our study area were not common, as opposed to a survey conducted by [30] in the neighboring Khorezm province where households cultivated 17 different, simultaneous tree-crop systems that mainly consisted of fruit species. The reason behind the sparse fruit orchards and trees in Karakalpakstan is the harsh local environmental conditions. Particularly, fruit orchards and trees are being either dried out during water shortage periods or water abundant periods, due to an increase in the level of salty groundwater [31]. As an intervention strategy for Karakalpakstan, it would be possible to integrate annual crops into tree-crop systems and have high returns, as was observed by [30] in similar conditions with cereal (47%), vegetable (27%), fodder (19%) and cash crops (7%) as the most prioritized crops. Estimations by [31] showed that three quarters of surveyed households have arable land to grow fruits and vegetables in Karakalpakstan. Here, it is also important to note that all arable lands and their conditions to access irrigation water are not the same [32]. When applying tree-crop system into the study site, it is vital to take into account the condition of land and water resources, as evidenced by [19]; there is a negative impact of the condition of land and water resources on the effective use of the available crop and livestock potential.

Livestock numbers, particularly cattle, are the most essential asset in rural dwellings. The author of [33] noted that cattle tend to be acquired later in life after childbearing is completed and households have acquired sufficient funds to invest in cattle. These observations, combined with factors such as large shares of non-agricultural income, indicate the need for household members to have entrepreneurial capacities, as well as opportunities for such capacities to be employed. As evidenced from the neighboring Khorezm province, [17] estimated that entrepreneurial capacities could be significant if all incomes from unrecorded or in-kind incomes could be estimated. At present, without accounting for informal or in-kind income sources and considering the lack of a private sector in rural areas of Uzbekistan, formal sectors supported by the state will remain as crucial sources of cash income, and agricultural activities will remain as mostly in-kind sources of income in Karakalpakstan. To include in-kind income in surveys, easy-to-use assessment approaches must be used so that households can associate monetary value to such types of income. This would provide a more accurate assessment of the financial assets of these households.

### 4.4. Implications for Rural Development Policy

From this study's findings as discussed above, in order to increase the livelihood diversification of smallholder systems in the ASB, which was used as an example for increasing the adaptive capacities of agrarian communities to climate and other unexpected changes, national and regional policy-makers and rural developers should consider promoting the following points: (1) recognizing the importance of non-agricultural activities

generating and diversifying household incomes across all types of smallholder systems in the areas, and that supporting the development of non-agricultural livelihood activities is also relevant because the components in household livelihood portfolios seem increasingly preferred by farmers; (2) encouraging government programs to create employment and protect the households socially, especially young families and other vulnerable groups; and (3) more importantly, supporting the development of a private sector would create new opportunities for the entrepreneurial capacities of the households.

## 5. Conclusions

In the presence of climatic variability and the risks involved with agricultural production for the rural people in the Aral Sea basin, current research has employed the SLF to investigate the factors affecting agricultural livelihoods at the village level. The SLF could also determine type-specific factors of the production and livelihood diversifications of these households compared to treating all the households as a group.

Consequently, in two rural villages in the Karauzyak district this study identified three agricultural livelihood types that were significantly distinct from each other.

The first livelihood type was found to be rich in land per household member and cattle dominant. Households in this livelihood type possessed relatively higher landholdings per household member on average compared to other livelihood types. Additionally, households in this group bred more livestock, most of which was cattle. In addition to these factors, these households were also richer in housing assets and had higher social capital and better welfare. Watermelons and fodder crops were cultivated more by this type of household. Regarding crop diversification, households in this group were significantly more diverse in crop production. Within this group, the study also found that distance to food markets from households negatively influenced the households' decisions to diversify their crop productions. However, this information's significance was still statistically weak.

The second livelihood type in the Karauzyak district was relatively labor rich, the land per household member was poorer and the dependency ratio was lower. In this type of livelihood, households had bigger families and more labor. Despite the large family sizes, the dependency ratio was much lower than in other household types. Regarding crop production, these households mainly cultivate vegetables and, hence, less diversified crops. Moreover, the regression analysis revealed major drivers that influenced the households' decisions to diversify crop productions. Education was found to be one of the driving forces that positively affected diversification in this livelihood type, implying that households with educated members are predisposed to diversifying their crop productions. Additionally, higher shares of cattle in the total amount of livestock owned and the landholding sizes of the households positively impacted the diversification.

The last livelihood type had relatively young household heads, less labor, fewer household members, and incomes outside of agriculture. Households without higher education were more prevalent in this livelihood type. This livelihood type also had less diverse crop productions when compared to the second livelihood type. Regarding this livelihood type, the study found that the number of potential workers in the households and the number of cattle and small ruminants positively influenced the households' decisions to diversify crop production. However, distance to food markets negatively influenced the diversification.

Overall, our analysis of the total sampled populations found that the agricultural experiences of the household heads, households with educated members, shares of cattle, shares of on-farm income, landholding per household member, and distance to food markets were the drivers that influenced each household's decision to diversify its crop production.

**Author Contributions:** Conceptualization, A.A. (Akmal Akramkhanov) and Q.B.L.; Methodology, A.A. (Akmal Akramkhanov) and Q.B.L.; Software, A.A. (Akmal Akramkhanov) and A.A. (Adkham Akbarov); Validation, A.A. (Akmal Akramkhanov) and A.A. (Adkham Akbarov); Formal analysis, A.A. (Akmal Akramkhanov), A.A. (Adkham Akbarov) and Q.B.L.; Data curation, A.A. (Akmal Akramkhanov), A.A. (Adkham Akbarov) and Q.B.L.; Writing—original draft, A.A. (Ak-

mal Akramkhanov), A.A. (Adkham Akbarov) and Q.B.L.; Writing—review & editing, A.A. (Akmal Akramkhanov), S.U. and Q.B.L.; Visualization, Q.B.L.; Supervision, A.A. (Akmal Akramkhanov) and Q.B.L.; Project administration, A.A. (Akmal Akramkhanov). All authors have read and agreed to the published version of the manuscript.

**Funding:** This research was funded by the CGIAR Research Program on Dryland Systems led by the International Center for Agricultural Research in the Dry Areas (ICARDA). It was continued to be supported by the CGIAR Initiative on Sustainable Intensification for Mixed Farming Systems (SI-MFS). The opinions expressed here belong to the authors, and do not necessarily reflect those of CRP-DS, SI-MFS, ICARDA, or CGIAR.

**Data Availability Statement:** Dataset is available at the following link https://hdl.handle.net/20.5 00.11766/3495, accessed on 1 September 2022.

**Conflicts of Interest:** The authors declare no conflict of interest. The funders had no role in the design of the study; in the collection, analyses, or interpretation of data; in the writing of the manuscript, or in the decision to publish the results.

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
