# Peer review of "Agricultural Livelihood Types and Type-Specific Drivers of Crop Production Diversification: Evidence from Aral Sea Basin Region"

_sustainability, doi:10.3390/su15010065_

Round 1

Author Response

Dear Reviewer,

Thank you very much for your valuable comments and time!

Please, find the responses to your comments in the attached file.

Kind regards,

The authors

Reviewer 2 Report

The author's comprehensively elaborate the agricultural livelihood types and type specific drivers of production diversification. The methodology is sound and use of  sustainable livelihood framework is appropriate.

However, moderate langaugar corrections are needed. 

Author Response

Dear Reviewer,

Thank you very much for your time and comment.

Please, find the answers to your feedback in the attached file.

Kind regards,

the authors

Reviewer 3 Report

Thank you for the opportunity to learn from this work. I think that the empirical presentation is generally sound, and I really only have some more specific reservations about some details in the methods that I think should be cleared up. I'll list these below:

1) The manuscript says the 100-household sample comes from 1,384 selected households (line 137) - I think the meaning here is that this is the total population of households in the two villages. If so, please make that point clear. If not, please explain how this selection was accomplished.

2) I think the PCA results could do with more interpretation. First, components that are essentially just a single variable don't seem to tell us much. Second, this is a much larger number of retained components than you would typically see in this kind of analysis, so I think there might need to be some more discussion here about how you decided to set the threshold of explained variation that was selected. Third, the naming of the components seems to me a bit misleading. While I understand that focusing on the variable with the highest factor loading for each component is helpful, in several cases there are actually several variables with factor loadings near the highest level observed. That makes the interpretation of the component more complicated. This appears to be particularly the case for PC-4, which might be interpreted as capturing more general affluence, given the distribution of factor loadings found there.

3) k-means clustering requires the user to set the number of clusters at the outset. There should be some justification in the methods for why three clusters was selected - what, if any, comparisons were made with alternative numbers of clusters?

4) I have some pretty significant qualms about the regression analysis broken into clusters. First, the number of variables relative to number of observations is pretty high even for the data on all 100 households. This is even more of a squeeze for the clusters, which is going to inflate the standard errors. Now, on the one hand, that could mean that the variables you find to be statistically significant in each cluster are quite significant, but it also means that you are losing power to detect significant relationships, so null findings could just be because you don't have sufficient observations. Second, because the clusters are created using variables that are then included on both sides of the regression equations, it becomes very difficult to interpret the analysis. The thing is that the variables by which crop diversity is computed are also part of the analysis used to create the clusters. A third issue is that, because the clusters are by definition restricted to observations with a narrowed range of values on some of the variables, this will necessarily affect the magnitude of the estimated coefficients.

5) You need to be careful about using causal language in this analysis. While I'm usually not much of a stickler about this, in this case there are definitely some pretty big risks of reciprocal causality that need to be considered. Assuming that diversification does improve households' resilience, then that could potentially allow them to accrue more assets, which could in turn facilitate more diversity, for example. In that case, it becomes difficult to say what is causing what without longitudinal data. As a result, it is best in this case to keep the claims to the level of association only, and not wade into language like "drivers".

6) A minor point - I'm not clear on how/why those particular variables were selected to show the profiles in Figure 3. Perhaps a bit more explanation in the figure caption or the text would be helpful.

Author Response

Dear Reviewer, 

Thank you very much for your spending time and comments to improve the manuscript. Please, find the responses in the attached file.

Thank you very much!

The authors

Round 2

Reviewer 1 Report

The authors have addressed the issues raised in their revised manuscript and rebuttal letter.